# EFFECTIVE LLM KNOWLEDGE LEARNING REQUIRES RETHINKING GENERALIZATION

## ABSTRACT

Large language models (LLMs) are trained on a substantial amount of documents that contain extensive world knowledge. However, it is still not well-understood how knowledge is acquired via autoregressive pre-training and extracted via question-answering. This lack of understanding greatly hinders effective knowledge learning, especially for continued pre-training on up-to-date information, as this evolving information often does not have diverse repetitions like foundational knowledge. In this paper, we focus on understanding and improving LLM knowledge learning. We found and verified that knowledge learning for LLMs can be deemed as an implicit supervised task hidden in the autoregressive pre-training objective. Our findings suggest that knowledge learning for LLMs would benefit from methods designed to improve generalization ability for supervised tasks. Based on our analysis, we propose to diversify training documents' formats as data augmentation to grow in-distribution samples. This data augmentation method does not present the risk of altering the facts embedded in documents as text paraphrasing. We also introduce sharpness-aware minimization as an effective optimization algorithm to better improve generalization. Moreover, we adapt our method to instruction tuning for generalization to various phrasings of questions. Extensive experiment results validate our findings and demonstrate our methods' effectiveness in improving knowledge learning in both the continued pre-training and instruction tuning stages. This paper offers new perspectives and insights to interpret and design effective strategies for LLM knowledge learning.

## 1 INTRODUCTION

Large language models (LLMs) are pre-trained on large-scale datasets encompassing extensive knowledge, enabling them to demonstrate remarkable performance on knowledge-intensive tasks (Brown et al. (2020); OpenAI (2023); Chowdhery et al. (2023); Zhang et al. (2022); Touvron et al. (2023a;b); Gemini Team (2023); Yang et al. (2024)). Pre-trained LLMs are able to answer questions on various domains, like "Who was the first president of America?", or "What is the second highest mountain in the world?". However, it is still unclear how LLMs can acquire knowledge from the training corpus and answer these questions. This insufficient understanding severely limits the efficiency of knowledge acquisition, particularly when it comes to continued pre-training on newly updated information. Unlike foundational or textbook knowledge, which typically benefits from diverse repetitions and widespread coverage, evolving information often lacks such extensive variation, making it more challenging for LLMs to learn (Kandpal et al. (2023); Jiang et al. (2024); Allen-Zhu & Li (2024)). Therefore, we hope to better understand the mechanism for autoregressive language model knowledge learning and enhance it.

We approach this problem by carefully examining the training objective and inference formula for autoregressive language models. Assuming pre-training on the document $d =$ "*Elon Musk was raised in* ***South Africa***.", people might ask questions querying knowledge stored in the bold tokens, which we refer to as knowledge tokens. For example, successful elicitation of Elon Musk's hometown via question-answering is possible only if $P(South|tokens\ preceding\ South\ in\ q)$ and $P(Africa|tokens\ preceding\ Africa\ in\ q)$ are high enough, where $q =$ "*Question : Where did Elon Musk grow up? Answer : South Africa*". Then we analyze how autoregressive pre-training on documents increases this conditional probability. The training objective of autoregressive language models can be formally

interpreted as minimizing the negative log-likelihood (NLL) loss over a dataset, where each sample is constructed such that each token in a document serves as the target label, and its preceding tokens are used as the input context. Therefore, we hypothesize document training samples and questions that share the same knowledge tokens as labels come from the same distribution. In this way, minimizing NLL loss for document samples would generalize to questions. In the example $d$ and $q$ above, it means that increasing $P(South|tokens\ preceding\ South\ in\ d)$ and $P(Africa|tokens\ preceding\ Africa\ in\ d)$ would generalize and thereby increases $P(South|tokens\ preceding\ South\ in\ q)$ and $P(Africa|tokens\ preceding\ Africa\ in\ q)$. However, this is not obvious from the human perspective, as document training samples and questions differ in structure: one is a declarative sentence, while the other is in the form of QA. Therefore, we construct a human biography dataset including different attributes similar to Allen-Zhu & Li (2024). During training, we observe that the accuracy of predicting attribute knowledge tokens given questions as inputs increases along with the accuracy conditioned on the training document sequence. We also observe that integrating paraphrased documents in training can further improve the accuracy of question answering. These observations verifies our hypothesis that document training samples and questions that share the same knowledge token labels come from the same distribution and thus LLM knowledge learning is implicitly a supervised problem.

As we have demonstrated that LLM knowledge learning is supervised, we then explore methods to improve the generalization ability. According to our perspective, with a single document demonstrating the knowledge, LLM knowledge learning is a 1-shot supervised learning problem. With paraphrased documents, LLM knowledge learning is a few-shot supervised learning problem. Therefore, the first and foremost task is to increase the number of in-distribution samples since generalization for few-shot learning is extremely difficult. However, reliable rephrasing documents by humans is expensive and time-consuming, while LLM rephrasing might alter facts in documents (Ding et al. (2024)). Witnessing texts can be presented in different formatting without affecting embedded facts, we propose modifying documents' formats as data augmentations. By presenting the same training document with different spacing or padding, the number of in-distribution samples for knowledge learning is increased. Then, with adequate in-distribution samples synthesized using our data augmentation, we utilize sharpness-aware minimization (SAM, Foret et al. (2021)) as the optimization method to better improve generalization. Last, we identify the importance of generalizing to diverse questions that share the same answer, an aspect critical for effective knowledge extraction but overlooked in instruction tuning (Wang et al. (2024); Bukharin & Zhao (2024)). Thus we also integrate our methods into the instruction tuning phase to enhance the generalization ability for question-answering. Specifically, we use SAM as the optimization method and augment the questions of the training QA pairs. In this way, the instruction-tuned model can respond more accurately to different rephrasings of the questions in the training QA pairs, and apply analogous QA patterns for similar questions about all pre-training documents.

We evaluate our methods on our constructed biography dataset similar to Allen-Zhu & Li (2024), and the Wiki2023 dataset (Jiang et al. (2024)). Results show that our approach leads to nontrivial improvement of generalization ability in both the continued pre-training and instruction tuning phases. In addition, detailed ablations validate our finding that generalization matters for LLM knowledge acquisition and extraction. The main contributions of this paper are summarized as follows:

- We hypothesize and verify that LLM knowledge learning is implicitly a supervised learning problem. This novel perspective provides a solid foundation and systematic way for analyzing and improving knowledge learning ability.

- To improve the generalization ability for LLM knowledge learning, we propose to generate in-distribution training samples by diverse document formatting. This automatic augmentation method mitigates the risk of altering facts in documents, in contrast to rephrasing. We further apply SAM as the optimizer to enhance the generalization ability.

- We point out the importance of generalization on different questions that share the same answer, an aspect critical for effective knowledge extraction but previously neglected in the instruction tuning phase. We use SAM and apply our data augmentation to the question part of QA pairs used in instruction tuning, to enhance the knowledge extraction ability.

- Extensive experimental results and ablation studies validate the supervised nature of LLM knowledge learning and demonstrate our methods' effectiveness in improving knowledge learning in both the continued pre-training and instruction tuning phases.

## 2 RELATED WORK

**Understanding of LLM Knowledge Learning.** There are several works trying to understand how LLMs learn knowledge from documents and retrieve them in question answering. Akyürek et al. (2022) tries to detect training documents important for question-answering for pre-trained LLMs. A number of works find connections between the frequency of certain knowledge appearing in pre-training documents and its question-answering ability (Kandpal et al. (2023); Akyürek et al. (2022); Petroni et al. (2019); Kassner et al. (2020); Wei et al. (2021); Févry et al. (2020); De Cao et al. (2021)). Recently, Allen-Zhu & Li (2024) and Ovadia et al. (2024) empirically observe that adding rephrased documents describing the same knowledge in the pre-training phase helps knowledge extraction after conducting instruction tuning. These works primarily try to explain LLM knowledge learning by summarizing observed patterns and their analysis is only confined to LLMs after the instruction tuning stage. Our work, on the other hand, provides a more rigorous and systematic explanation. Our analysis covers both the continued pre-training and instruction tuning stages.

**Continued LLM Knowledge Learning.** As the pre-trained knowledge stored in LLMs quickly becomes outdated, adapting up-to-date information into LLMs becomes a critical problem. The primary approach to tackling this problem is through continued pre-training on documents containing up-to-date knowledge (Ovadia et al. (2024); Jiang et al. (2024); Jang et al. (2022)). However, straightforward autoregressive pre-training on new corpus usually cannot lead to effective knowledge acquisition. This is likely due to the lack of diverse demonstrations of the same knowledge like foundational or textbook knowledge (Allen-Zhu & Li (2024); Jiang et al. (2024); Ovadia et al. (2024); Cheng et al. (2024)). Therefore, some works focus on rephrasing documents to alleviate this issue (Cheng et al. (2024); Allen-Zhu & Li (2024); Ovadia et al. (2024)). However, paraphrasing documents manually can be expensive and tedious, while paraphrasing by LLM might not be reliable as facts and knowledge inside documents could be changed in this process (Ding et al. (2024)). Therefore, we aim to avoid the risk of changing facts embedded in documents while enabling effective knowledge acquisition. Another line of work tries to include QA data together with or before adapting to new documents (Allen-Zhu & Li (2024); Jiang et al. (2024)). However, these methods introduce new difficulties in finding effective arrangements and proportions of QA data and documents. To induce effective knowledge extraction during inference, instruction tuning on annotated QA pairs after training on raw documents has recently become a common practice (Sanh et al. (2022); Wei et al. (2022); Mishra et al. (2022); Iyer et al. (2022); Kopf et al. (2023); Zhou et al. (2023); Sun et al. (2023b;a)). Current instruction tuning methods generally focus on diversifying the domain of QA pairs so that questions from different areas can be answered after instruction tuning (Bukharin & Zhao (2024); Wang et al. (2024)). However, the diversity of questions with the same answer is largely overlooked. We identify that different users might pose the same question using different wordings and phrases, and therefore propose to also take this into consideration for instruction tuning.

**Data Augmentation for Natural Language Processing.** There is a rich literature on data augmentation techniques used for natural language processing (Chen et al. (2021); Ding et al. (2024); Wei & Zou (2019)). A popular type of traditional data augmentation method is synonym substitution, which replaces words in documents with their synonyms according to some pre-defined dictionaries (Kolomiyets et al. (2011); Yang (2015); Zhang et al. (2015)). Another popular class of traditional data augmentation method is inserting, replacing, deleting, and swapping words in documents (Wei & Zou (2019); Iyer et al. (2022); Niu & Bansal (2018); Miao et al. (2020)). In the era of LLMs, paraphrasing documents or synthesizing data using LLMs become increasingly popular (Ding et al. (2024); Sharma et al. (2023); Nair et al. (2023)). However, these data augmentation methods generally modify the semantics of original documents, and some tokens where the knowledge and facts reside have the risk of being altered (Ding et al. (2024)). Therefore, we opt to avoid such risk and design our data augmentation method for LLM knowledge learning.

## 3 LLM KNOWLEDGE LEARNING AS SUPERVISED LEARNING

### 3.1 AUTOREGRESSIVE LANGUAGE MODEL

Let $\boldsymbol{\theta}$ denote the parameters of an autoregressive language model, and $\mathcal{V}$ be a fixed vocabulary of tokens. Suppose document $\boldsymbol{d}$ contains a sequence of tokens $(x_1, x_2, \ldots, x_T)$ from $\mathcal{V}$, where $T$ is

the length of the document. The training sequence has a special token $x_0$ prepended, indicating the sequence's start. The autoregressive language modeling task estimates the conditional probability distribution $P(x_t|x_{<t})$ for each $t = 1, 2, \ldots, T$. This is typically achieved by training a deep neural network to predict the next token $x_t$ given the previous tokens $x_0, x_1, x_2, \ldots, x_{t-1}$. For document $\boldsymbol{d}$, the negative log-likelihood loss function of the observed sequence $\boldsymbol{d}$ is:

$$\ell(\boldsymbol{\theta}, \boldsymbol{d}) = -\log P_{\boldsymbol{\theta}}(\boldsymbol{d}) = -\log \prod_{t=1}^{T} P_{\boldsymbol{\theta}}(x_t|x_{<t}) = -\sum_{t=1}^{T} \log P_{\boldsymbol{\theta}}(x_t|x_{<t}). \tag{1}$$

During inference, given a sequence of $m$ tokens $(x_1, \ldots, x_m)$ as context, the autoregressive language model iteratively generates the next $n$ tokens with one token at a time conditioned on the context and prior generations to obtain the completed sequence $(x_1, \ldots, x_{m+n})$:

$$\log P_{\boldsymbol{\theta}}(x_{m+1} \ldots x_{m+n}|x_{<m+1}) = \log \prod_{t=m+1}^{m+n} P_{\boldsymbol{\theta}}(x_t|x_{<t}) = \sum_{t=m+1}^{m+n} \log P_{\boldsymbol{\theta}}(x_t|x_{<t}). \tag{2}$$

Generation at each step would either greedily select the token with the highest likelihood or use sampling methods such as top-$k$ and nucleus sampling (Holtzman et al. (2020); Fan et al. (2018); Ippolito et al. (2019)).

### 3.2 CASTING KNOWLEDGE LEARNING AS A SUPERVISED LEARNING PROBLEM

Assuming a language model is trained on a document $\boldsymbol{d}$ = *"Elon Musk was born on June 28, 1971. He was raised in South Africa."*, users would often be interested in asking questions querying knowledge stored in the bold tokens, which we refer to as **knowledge tokens**. From Eq. 2, we can see that these questions can be answered either by prompting (Brown et al. (2020); Petroni et al. (2019); Roberts et al. (2020)) or direct question-answering after instruction tuning (Sanh et al. (2022); Wei et al. (2022); Ouyang et al. (2022)) only if the probabilities of knowledge tokens conditioned on context questions are high enough. For example, Elon Musk's hometown can be reliably elicited from a language model trained on $\boldsymbol{d}$ only if $P(South|tokens\ preceding\ South\ in\ \boldsymbol{q})$ and $P(Africa|tokens\ preceding\ Africa\ in\ \boldsymbol{q})$ are high enough, where $\boldsymbol{q}$ = *"Question : Where did Elon Musk grow up? Answer : South Africa"*. Next, we analyze how autoregressive pre-training on document $\boldsymbol{d}$ increases these knowledge tokens' conditional probabilities.

For autoregressive language model training on document $\boldsymbol{d}$, we can regard $x_{i+1}$ as the pseudo-label $y_i$ of the input training sequence $(x_0, x_1, x_2, \ldots, x_i)$, i.e., $y_i = x_{i+1}$, which results in a training sample $((x_0, x_1, \ldots, x_i); y_i)$. Thus, the document $\boldsymbol{d}$ can be deemed as a set of training samples $\mathcal{S}(\boldsymbol{d}) = \{((x_0); y_0), ((x_0, x_1); y_1), ((x_0, x_1, x_2); y_2), \ldots, ((x_0, x_1, \ldots, x_{T-1}); y_{T-1})\}$. The language model autoregressively trained on $\mathcal{S}(\boldsymbol{d})$ minimizes the negative log-likelihood (NLL) loss for each training sample. Furthermore, given a set of training documents $\mathbb{D}$, the training loss along with the training objective is:

$$\min_{\boldsymbol{\theta}} L_{\mathbb{D}}(\boldsymbol{\theta}) = \frac{1}{|\mathbb{D}|} \sum_{\boldsymbol{d} \in \mathbb{D}} \frac{1}{|\mathcal{S}(d)|} \sum_{\boldsymbol{r} \in \mathcal{S}(\boldsymbol{d})} [-\log P_{\boldsymbol{\theta}}(\boldsymbol{r})]. \tag{3}$$

Then, for training samples $((x_0, x_1, \ldots, x_i); y_i)$ with knowledge tokens as labels, **we hypothesize questions that share the same knowledge tokens as labels are from the same distribution of the training samples.** If this is the case, then minimizing the NLL loss for training samples would generalize and increase corresponding knowledge tokens' probabilities conditioned on input question prompts. Taking the $\boldsymbol{d}$ and $\boldsymbol{q}$ above as an example, it means that increasing $P(South|tokens\ preceding\ South\ in\ \boldsymbol{d})$ and $P(Africa|tokens\ preceding\ Africa\ in\ \boldsymbol{d})$ would generalize and thereby increases $P(South|tokens\ preceding\ South\ in\ \boldsymbol{q})$ and $P(Africa|tokens\ preceding\ Africa\ in\ \boldsymbol{q})$. Thereafter, knowledge learning for LLM can be considered as a supervised learning problem. However, this is not obvious from the human perspective, as training samples and their corresponding questions are quite different in the input space: one is a declarative sentence, while the other is in the form of QA. For example, although sharing "South" as the label, humans can hardly consider "Question: Where did Elon Musk grow up? Answer:" and "Elon Musk was born on June 28, 1971. He was raised in" being from the same distribution. Therefore, we need to verify this hypothesis first.

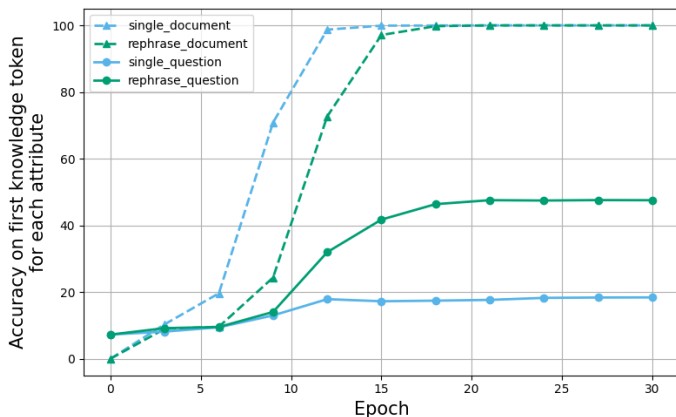

Figure 1: Average first knowledge token accuracy for each attribute conditioned on: (1) context training document sentences, (2) context questions.

### 3.3 VERIFICATION

To validate our hypothesis in a controlled setting, we generate synthetic human biography data similar to Allen-Zhu & Li (2024). Specifically, we create 1,000 randomly generated human biography profiles, each characterized by five attributes: birth date, college, major, hometown, and company. To construct training documents, we employ a predefined template to represent these five attributes and leverage LLM to generate two additional paraphrased templates. Each profile's attributes are populated into these templates to form the training dataset. For evaluation, we generate a testing dataset by creating five diverse questions for each attribute of an individual, ensuring question variety. Additionally, we produce one QA pair per attribute for each profile to support instruction fine-tuning. Below, we provide an example of a biography text entry along with a QA pair:

- Eden Benitez completed his education at University of Wisconsin, Madison. His field of study was Marketing. He was employed at General Dynamics. His place of origin was Santa Clarita. He entered the world on January 18, 1959.

- Question: Which company did Eden Benitez have a professional role at? Answer: General Dynamics.

Then we study two scenarios on whether training on raw documents can generalize to question-answering: (1) **single**, training on biography documents where each biography profile is filled into a single template to form a single text entry; (2) **rephrase**, training on biography documents where each biography profile is filled into three different templates to form three text entries. We continually pre-train from Qwen 2 1.5B model (Yang et al. (2024)) with the above setting and record the **first**[1] knowledge token's accuracy conditioned on sentences from training documents and testing questions[2]. If $P(first\_knowledge\_token|tokens\ preceding\ first\_knowledge\_token\ in\ question)$ increases along with $P(first\_knowledge\_token|tokens\ preceding\ first\_knowledge\_token\ in\ document)$, LLM knowledge learning should be considered as a supervised problem. When we consider the biography above, $first\_knowledge\_token$ would be "General", $tokens\ preceding$ $first\_knowledge\_token\ in\ question$ would be "Question: Which company did Eden Benitez have a professional role at? Answer:", and $tokens\ preceding\ first\_knowledge\_token\ in\ document$ would be "Eden Benitez completed his education at University of Wisconsin, Madison. His field of study was Marketing. He was employed at".

In Fig. 1, dashed lines represent the accuracy of first knowledge tokens conditioned on sentences from training documents, while solid lines represent the accuracy of first knowledge tokens condi-

---

[1]The accuracy of all knowledge tokens (exact match) demonstrates similar trends and is reported in the experimental section.

[2]To align with the QA format that directly outputs knowledge tokens, a held-out QA pair is prepended to testing questions.

tioned on testing questions. We can see that the testing accuracy on questions is increasing along with the training accuracy conditioned on documents. Moreover, we can see that training on all rephrased biography text entries leads to much higher accuracy on questions than training on a single text entry. These observations verify our hypothesis that input document training samples and questions that share the same knowledge token labels come from the same distribution and thus LLM knowledge learning is implicitly a supervised problem. From our perspective, when a single document is used to demonstrate the knowledge, LLM knowledge learning can be characterized as a 1-shot supervised learning problem. In contrast, when paraphrased documents are provided, it transitions into a few-shot supervised learning scenario. Insufficient in-distribution training samples hinder the model's ability to effectively acquire knowledge.

## 4 METHODS

Sec. 3 has posed LLM knowledge learning as a supervised learning problem. This section explores methods to improve the generalization ability for LLM knowledge learning. As knowledge tokens of a document are usually unknown, the methods developed are applied to all tokens in the document.

### 4.1 DATA AUGMENTATION VIA FORMATTING

As we analyzed in Sec. 3.3, knowledge learning without enough rephrased documents can be extremely difficult due to insufficient in-distribution samples with knowledge token labels. However, paraphrasing documents manually can be expensive and laborious, while paraphrasing using LLM might not be reliable. It is not easy to ensure that knowledge or facts in documents are not altered by LLM during paraphrasing (Ding et al. (2024)). Moreover, certain expressions and terminologies are irreplaceable and must be used in their exact form. Nor should mottoes and poems be rephrased when they are in training documents. Therefore, it is crucial to develop methods to reliably increase in-distribution samples with knowledge token labels without paraphrasing.

We may often encounter variations in the formatting used to present texts, such as whether to indent the beginning of a paragraph and whether to use spaces or tabs as indentations for codes. There are also variations for using single-space or double-space spacing in the era of typewriters (Wikipedia contributors (2024)). These formatting differences, while altering some of the format tokens, do not affect the semantic meaning and knowledge of the text itself. Therefore, given a training document $d$, we propose to apply the following formatting-based data augmentations:

- **Wrapping.** Augmented documents are created by wrapping document $d$ with quotes, asterisks, brackets, or parentheses. This is used to imitate the case that the document is quoted, highlighted, or appears in a Markdown document.
- **Left padding.** Augmented documents are created by padding spaces, tabs, or pound signs to the left of the document $d$. This is to mimic the scenarios of $d$ appearing in a document written using Markdown or as a paragraph in a paper.
- **Random space insertion.** Augmented documents are created by randomly inserting additional spaces adjacent to original spaces in $d$. This simulates the case that the training document is presented using different spacing and includes some unintentional extra spaces.

Therefore, for a training document being "Elon Musk was raised in South Africa.", some examples of its augmentations are as follows:

**Wrapping augmented examples**

```
"Elon Musk was raised in South Africa."
*Elon Musk was raised in South Africa.*
```

**Left padding augmented examples**

```
    Elon Musk was raised in South Africa.
# Elon Musk was raised in South Africa.
```

**Random space insertion augmented examples**

```
Elon  Musk was raised in South  Africa.
Elon Musk was raised in  South Africa.
```

Detailed specifications of the data augmentations are discussed in Appendix C.1. With these augmented documents, we diversify the in-distribution samples with knowledge token labels while not changing the knowledge and facts inside these documents.

## 4.2 SHARPNESS-AWARE MINIMIZATION

With our proposed data augmentation methods increasing in-distribution samples with knowledge token labels, we can further enhance the generalization ability by applying generalizable optimization or regularization methods designed for traditional supervised problems. Recently, Foret et al. (2021) developed the *Sharpness-Aware Minimization* (SAM) to improve the generalization ability of DNN for supervised problems, which has achieved substantial generalization improvement on widely studied supervised learning problems like image classification (Baek et al. (2024); Chen et al. (2022); Foret et al. (2021)). We adopt this technique for the LLM knowledge learning task.

Given a training document $d$, let $\mathcal{B} = \mathcal{S}(d)$, and according to SAM we solve the following problem:

$$\min_{\boldsymbol{\theta}} \max_{\|\boldsymbol{\epsilon}\|_2 \leq \rho} L_{\mathcal{B}}(\boldsymbol{\theta} + \boldsymbol{\epsilon}) + \lambda\|\boldsymbol{\theta}\|_2^2, \tag{4}$$

where $\rho \geq 0$ is a given perturbation radius, $\lambda$ is a small positive regularization constant. The objective is to find a minimizer with the neighborhood where the loss does not increase too much. According to SAM, the inner maximization problem in Eq. (4) is solved approximately at $\hat{\boldsymbol{\epsilon}} = \rho\nabla L_{\mathcal{B}}(\boldsymbol{\theta})/\|\nabla L_{\mathcal{B}}(\boldsymbol{\theta})\|_2$ by the first-order Taylor expansion. Then, the objective function of Eq. (4) changes to $L_{\mathcal{B}}(\boldsymbol{\theta} + \hat{\boldsymbol{\epsilon}}) + \lambda\|\boldsymbol{\theta}\|_2^2$, on which the gradient descent is performed.

## 4.3 ADAPTATION TO INSTRUCTION TUNING

Instruction tuning has recently become a common practice to make LLMs follow human instructions and perform question-answering (Sanh et al. (2022); Wei et al. (2022); Ouyang et al. (2022)). Instruction tuning computes the negative log-likelihood loss only on tokens in the answer with the question as the context: $L_{\boldsymbol{a}} = -\sum_t \log P(\boldsymbol{a}_t|\boldsymbol{q}, \boldsymbol{a}_{<t})$. The QA pairs used in instruction tuning are derived from specific documents in the pre-training dataset. After instruction tuning, LLM can make analogies and perform similar QAs on other documents seen during the pre-training phase. We identify that different users might pose variations of the same question to LLM, thus generalization on diverse questions sharing the same answer is crucial. Therefore, we use SAM and apply our data augmentation only to the context questions for instruction tuning. In this way, LLM would be able to respond accurately to different rephrases of a question seen during instruction tuning, consistently eliciting the same correct answer. As instruction tuning would make LLM apply analogous QA patterns for other documents seen during pre-training, we expect the generalization ability can also be brought to other pre-training documents. As prior works on instruction tuning generally focus on the diversity of QA pairs from different domains and tasks while ignoring the diversity of questions with the same answer (Bukharin & Zhao (2024); Wang et al. (2024)), we hope our exploration can bring more insights.

## 5 EXPERIMENTS

### 5.1 EXPERIMENT SETTINGS

**Baseline methods.** We experiment with two standard baselines: (1) continued pre-training and (2) continued pre-training with instruction tuning (SFT), and demonstrate the effectiveness of our methods in improving knowledge learning abilities of these baselines.

**Base models.** We use Qwen 2 1.5B (Yang et al. (2024)) and LLaMA 2 7B (Touvron et al. (2023b)) as base models and test all baselines and their combination with our methods on these models.

**Datasets.** We use our generated biography dataset and Wiki2023-film dataset proposed in Jiang et al. (2024) for the experiment. For the biography dataset, we follow Allen-Zhu & Li (2024) to continually pre-train on all individuals and instruction-tune on 1 QA pair per attribute of half of the individuals. Our evaluation differs from Allen-Zhu & Li (2024), which evaluates only 1 question prompt that uses the same template as the one used for instruction tuning, for the remaining half individuals. We generate 5 different question prompts for each attribute to better evaluate the

generalization ability, totaling 12500 QA pairs. The Wiki2023-film dataset we used is regenerated following the same recipe in Jiang et al. (2024) since the original dataset is not publicly available. The biography dataset is synthetic while the recipe for generating the Wiki2023-film dataset tries to minimize overlap with the pre-training corpus. Thus, experimenting on these two datasets can mimic the difficult case of continued knowledge learning on up-to-date information.

**Hyperparameter settings.** We use AdamW (Loshchilov & Hutter (2019)) as the base optimizer and a weight decay of 0.1. The learning rate for continued pre-training is set to 3e-5 while the learning rate for instruction tuning is set to 5e-6 for experiments on both the biography and Wiki2023 dataset. We use a batch size of 128 for the biography dataset and a batch size of 256 for the Wiki2023 dataset. For continued pre-training, we include the value of $\rho$ in Tab. 1. For instruction tuning, we use $\rho = 0.025$ for all our experiments. For the experiment on the Wiki2023 dataset (Jiang et al. (2024)), we continually pre-train both Qwen 2 1.5B (Yang et al. (2024)) and LLaMA 2 7B (Touvron et al. (2023b)) for 30 epochs. Qwen 2 1.5B is instruction-tuned for 5 epochs while LLaMA 2 7B is tuned for 2 epochs. For the experiment on the biography dataset, we continually pre-train both Qwen 2 1.5B (Yang et al. (2024)) for 30 epochs and LLaMA 2 7B (Touvron et al. (2023b)) for 15 epochs. Both models are instruction-tuned for 5 epochs. We use the iteration number of training without rephrased samples as a reference and ensure that all methods, regardless of data augmentation or the addition of paraphrased texts, are training for the same number of iterations to ensure a fair comparison.

Table 1: The value of SAM's $\rho$ used in different continued pre-training experiment settings.

| Base model | Biography w/o rephrase | Biography w/ rephrase | Wiki2023-film |
|---|---|---|---|
| Qwen 2 1.5B | 0.05 | 0.015 | 0.05 |
| LLaMA 2 7B | 0.025 | 0.015 | 0.025 |

**Evaluation metrics.** As we aim to evaluate the closed-book free-form question-answering ability, we utilize exact match (EM) between the model generations and ground truth answers as the evaluation metric (Kwiatkowski et al. (2019)). We also report Recall and F1 scores to better assess questions with long answers. When evaluating models that have not been instruction-tuned, we prepend 1 QA pair for the biography dataset and 5 QA pairs for the Wiki2023-film dataset to make sure that models can follow the QA format.

## 5.2 MAIN RESULTS

This section gives the main results comparing our methods with baselines. Unless otherwise specified, we use **base** to refer to the base model, **single** to refer to continued pre-training using the single document, **rephrase** to refer to continued pre-training on all paraphrased documents, and **ours** to refer to using both our data augmentation and the sharpness-aware minimization. For the results on instruction tuning, **ours** refers to using our methods on both the continued pre-training and instruction tuning stages.

Table 2: Evaluation results on the biography dataset with the base models continued pre-training and instruction-tuning w/ and w/o our methods. Our methods lead to substantial generalization improvement for both phases and show outstanding knowledge learning abilities compared to baselines.

| | Qwen 2 1.5B | | | Qwen 2 1.5B w/ SFT | | | LLaMA 2 7B | | | LLaMA 2 7B w/ SFT | | |
|---|---|---|---|---|---|---|---|---|---|---|---|---|
| | EM | Recall | F1 | EM | Recall | F1 | EM | Recall | F1 | EM | Recall | F1 |
| **base** | 0.7 | 7.1 | 6.2 | - | - | - | 0.7 | 8.9 | 7.6 | - | - | - |
| **single** | 7.1 | 16.1 | 12.4 | 52.8 | 57.6 | 57.1 | 52.0 | 59.1 | 58.5 | 89.6 | 91.3 | 91.2 |
| **w/ ours** | **43.2** | **57.7** | **52.3** | **57.9** | **62.2** | **61.8** | **85.4** | **88.2** | **88.0** | **93.3** | **94.2** | **94.1** |
| **rephrase** | 24.9 | 45.4 | 35.1 | 54.5 | 59.9 | 59.6 | 54.3 | 70.6 | 63.6 | 94.2 | 95.9 | 95.9 |
| **w/ ours** | **74.9** | **80.4** | **80.0** | **75.3** | **77.2** | **76.9** | **89.2** | **93.1** | **92.8** | **98.4** | **99.0** | **99.0** |

**Results on biography dataset.** We present the results on the biography dataset in Tab. 2. From Tab. 2, we can see that the base models cannot answer questions about the synthesized biography

profiles at all. This effectively simulates the case of LLMs adapting to up-to-date information, which is considered nontrivial (Jiang et al. (2024); Ovadia et al. (2024)). From the table, we can see that our methods lead to substantial improvement in knowledge learning when training on both non-rephrased and rephrased documents. Using our methods on non-rephrased documents significantly outperforms training on rephrased documents without our method during the continued pre-training phase, and leads to on-par performance after the instruction tuning phase. This result shows that our method can serve as an effective and reliable alternative to tedious and expensive manual rephrasing and unreliable LLM rephrasing. It can also be applied to documents containing mottoes or poems, which are not suitable for rephrasing. When applying our methods to rephrased documents, the knowledge learning performance becomes even better, showing that our methods can induce more generalization ability with more diverse in-distribution samples. This also demonstrates that our method can gain more enhancement when used together with paraphrasing. Moreover, we can see our methods lead to much more effective knowledge learning and extraction than baselines prior to the instruction tuning stage. The ability to extract learned knowledge at this early stage further demonstrates the effectiveness of our method in knowledge learning compared to rephrasing. This property could be beneficial in scenarios with limited resources, such as adapting LLMs to new domains where it is challenging and labor-intensive to annotate instruction-following examples.

Table 3: Evaluation results on the Wiki2023-film dataset with the base models continued pre-training and instruction-tuning w/ and w/o our methods. Our methods lead to nontrivial improvement in knowledge acquisition and extraction for both phases compared to baselines.

| | Qwen 2 1.5B | | | Qwen 2 1.5B w/ SFT | | | LLaMA 2 7B | | | LLaMA 2 7B w/ SFT | | |
|---|---|---|---|---|---|---|---|---|---|---|---|---|
| | EM | Recall | F1 | EM | Recall | F1 | EM | Recall | F1 | EM | Recall | F1 |
| base | 3.4 | 7.2 | 7.5 | - | - | - | 5.6 | 18.8 | 16.9 | - | - | - |
| single | 7.2 | 19.4 | 17.8 | 12.3 | 23.9 | 24.0 | 11.8 | 32.7 | 27.1 | 31.3 | 47.4 | 46.9 |
| w/ ours | 9.8 | 24.9 | 22.0 | 14.8 | 27.0 | 27.3 | 17.6 | 42.3 | 34.7 | 38.6 | 56.0 | 55.2 |

**Results on Wiki2023 dataset.** Next, we evaluate our methods with baselines on the Wiki2023-film dataset. As this dataset does not have rephrased training documents, we continually pre-train using a single document for all comparing methods. From Tab. 3 we can see that our methods lead to stable improvement over the baselines for both the continued pre-training and instruction tuning stages.

We can observe from Tab. 2 and Tab. 3 that our approach is consistently effective across different models, training phases, and datasets, demonstrating the robustness of our approach. We also want to stress that all comparing methods are trained with the same number of iterations in both the continued pre-training and instruction tuning phases. The performance gain of our approach and adding rephrased samples is not attributable to an increased number of training steps on enlarged datasets. On the other hand, it is because that generalization matters for LLM knowledge learning.

## 5.3 ABLATION STUDIES

In this section, we conduct comprehensive ablation studies on the effect of each component of our methods on both the continued pre-training and instruction tuning phases.

Table 4: Ablation study on the effect of integrating each component of our method into the continued pre-training phase.

| Training setting | Single | | | Rephrase | | |
|---|---|---|---|---|---|---|
| | EM | Recall | F1 | EM | Recall | F1 |
| Continued pre-train | 7.1 | 16.1 | 12.4 | 24.9 | 45.4 | 35.1 |
| w/ Data augmentation | 24.3 | 38.6 | 33.1 | 54.9 | 78.8 | 65.3 |
| w/ SAM | 19.7 | 29.1 | 26.6 | 52.8 | 63.3 | 60.8 |
| w/ Data augmentation + SAM | 43.2 | 57.7 | 52.3 | 74.9 | 80.4 | 80.0 |

**Effect of our methods on continued pre-training.** We first ablate the effect of our data augmentation and SAM for the continued pre-training phase. We use Qwen 2 1.5B (Yang et al. (2024)) as the

base model and conduct experiments on our synthesized biography dataset. We can see from Tab. 4 that when training with rephrased documents, both SAM and our data augmentation alone can bring measurable enhancement over the baseline. Furthermore, when SAM and our data augmentation are combined, the performance gains are further amplified. When training under the single document setting, the lack of in-distribution samples for knowledge tokens decreases the performance gain from SAM alone. Our data augmentation, on the other hand, brings adequate in-distribution samples, which leads to substantial improvement over the baseline. With these in-distribution samples, SAM is able to boost the performance even further.

Table 5: Ablation study on the effect of integrating each component of our method into the instruction tuning phase.

| Training setting | Pre-trained on **single** | | | Pre-trained on **rephrase** | | |
|---|---|---|---|---|---|---|
| | **EM** | **Recall** | **F1** | **EM** | **Recall** | **F1** |
| Instruction tuning | 52.8 | 57.6 | 57.1 | 70.4 | 72.8 | 72.6 |
| w/ Data augmentation | 55.3 | 59.5 | 59.2 | 73.2 | 75.1 | 74.9 |
| w/ SAM | 56.0 | 60.5 | 60.0 | 73.6 | 75.8 | 75.5 |
| w/ Data augmentation + SAM | 57.9 | 62.2 | 61.8 | 75.3 | 77.2 | 76.9 |

**Effect of our methods on instruction tuning.** Next, we ablate the effect of our methods on the instruction tuning phase. Still, we conduct experiments on our generated biography dataset. We use continual pre-trained Qwen 2 1.5B (Yang et al. (2024)) by our methods as the base model for instruction tuning. Prior works generally consider that knowledge is learned during continued pre-training and then made extractable in the instruction tuning phase (Allen-Zhu & Li (2024); Ouyang et al. (2022); Sanh et al. (2022); Wei et al. (2022)). Therefore, starting from the same continual pre-trained model, we can analyze how our methods influence knowledge extraction in this ablation. From Tab. 5, we can see that both SAM and data augmentation alone can improve knowledge elicitation over baseline instruction tuning. Furthermore, the combination of them leads to better performance. This result echoes our analysis in Sec. 4.3 that the generalization for questions with the same answer is crucial for effective and robust knowledge extraction and instruction following. Prior works generally focus on diversifying different QA pairs from different domains (Wang et al. (2024); Bukharin & Zhao (2024)), while ignoring this issue. We hope our analysis can provide a deeper understanding.

## 6 CONCLUSION

In this paper, we try to understand how LLMs acquire knowledge through autoregressive pre-training and retrieve the knowledge in question-answering. We found and verified that the knowledge learning for LLM is an implicitly supervised problem. We found that for certain knowledge tokens in documents that might serve as answers to questions, minimizing the negative log-likelihood loss on samples with prefixed document sequences as input and the next knowledge tokens as labels would generalize to samples with questions as input and the same knowledge token as labels. Thus, we verified that knowledge learning for LLMs is indeed a supervised problem. We subsequently propose a formatting-based data augmentation method to increase in-distribution samples via presenting training documents in different formats, which does not have the risk of altering knowledge and facts embedded in documents as paraphrasing. We also introduce the sharpness-aware minimization as the optimizer to better improve generalization ability. Then we extend our analysis to the instruction tuning phase and point out the importance of generalization on different questions with the same answer for effective knowledge extraction, which is overlooked by previous works. Extensive experiments and ablation studies validate our finding of the supervised nature of LLM knowledge learning and demonstrate our methods' effectiveness in improving knowledge acquisition and extraction for both continued pre-training and instruction tuning phases. We hope our work can provide insights to better understand and develop effective methods for LLM knowledge learning.

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

## A MORE EXPERIMENTS

### A.1 ABLATION ON DATA AUGMENTATION

We conduct the ablation study on the effect of our proposed three formatting-based augmentations for Qwen 2 1.5B on the biography dataset during the continued pre-training stage. The results are

Table 6: Ablation study on the effect of our proposed three types of data augmentation on the continued pre-training phase.

| Data Augmentation | EM | Recall | F1 |
|---|---|---|---|
| w/o data augmentation | 7.1 | 16.1 | 12.4 |
| w/ Wrapping | 21.0 | 34.0 | 28.3 |
| w/ Left padding | 10.7 | 24.1 | 16.9 |
| w/ Random space insertion | 15.2 | 37.7 | 24.7 |
| w/ all three data augmentation | 24.3 | 38.6 | 33.1 |

summarized in Tab. 6. It can be seen that all three types of formatting-based augmentation lead to significant improvement over the baseline. Among them, Wrapping and Random space insertion are more effective than Left padding. The combination of all three types of augmentation creates more diverse in-distribution samples and leads to more balanced considerable enhancement.

## A.2 COMPARISON WITH TRADITIONAL NLP AUGMENTATION

In this section, we compare our formatting-based data augmentation with a representative traditional NLP data augmentation technique, EDA (Wei & Zou (2019)). The experiment is conducted by continually pre-training Qwen 2 1.5B on the biography dataset. From Tab. 7, we can see that EDA is harmful for knowledge learning. The exact match decreases from 7.1 to 0 when applying EDA. This is because EDA uses random word insertion, random word deletion, and random word swap, which are highly likely to alter the knowledge in documents. This further demonstrates our formatting-based augmentation's advantage that it reliably increases the number of in-distribution samples without changing the knowledge in documents.

Table 7: Comparison between our formatting-based data augmentation and EDA (Wei & Zou (2019)).

| Data Augmentation | EM | Recall | F1 |
|---|---|---|---|
| w/o data augmentation | 7.1 | 16.1 | 12.4 |
| w/ EDA | 0.0 | 9.0 | 3.8 |
| w/ our data augmentation | 24.3 | 38.6 | 33.1 |

## A.3 MORE EXPERIMENTS OF PREDICTION ON FIRST KNOWLEDGE TOKEN

We compare the average first knowledge token accuracy conditioned on context questions for models continually pre-trained with and without our method in Fig. 2. The experiment setting is the same as in Sec. 3.3 of the main text. We can see that our method leads to significant improvement over naive continued pre-training with and without rephrased samples. This indicates our method's capability to improve the generalization ability for knowledge learning.

## B DISCUSSION ON THE REVERSAL CURSE

The reversal curse is an empirical observation that LLMs trained on "A is B" fail to learn "B is A" (Berglund et al. (2024)). We do not aim to alleviate the reversal curse problem in this paper, however, our proposed perspective to view LLM knowledge learning as a supervised learning problem can explain the existence of the reversal curse. Assume the training sentence is "<bos> A is B" and its reverse sentence is "<bos> B is A", where <bos> is the special begin-of-sentence token indicating the beginning of a sentence. During training, the input "<bos>" has "A" as the label. In the reverse sentence, the input "<bos> B is" has "A" as the label. "<bos>" and "<bos> B is" are too different

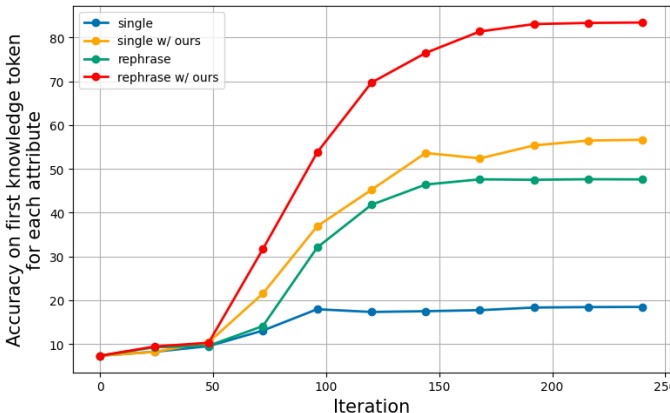

Figure 2: Comparison of average first knowledge token accuracy conditioned on context questions for models continually pre-trained with and without our method.

and are impossible to be from the same distribution. Although sharing "A" as the label, training on input "<bos>" cannot generalize to "<bos> B is". This explains the empirical observation of the reversal curse. This also corresponds to the empirical observation that paraphrasing the training sentence "<bos> A is B" cannot alleviate the reversal curse (Berglund et al. (2024)), and only reverse parts or the whole training sentence can help (Golovneva et al. (2024)).

## C  EXPERIMENT DETAILS

### C.1  DATA AUGMENTATION SPECIFICATIONS

We include all variations of data augmentations we used in the following. For the random space insertion augmentation, we randomly insert an additional space adjacent to spaces in documents with a probability of 0.2.

---

**Wrapping augmented examples**

```
'Elon Musk was raised in South Africa.'
"Elon Musk was raised in South Africa."
*Elon Musk was raised in South Africa.*
**Elon Musk was raised in South Africa.**
***Elon Musk was raised in South Africa.***
==Elon Musk was raised in South Africa.==
<Elon Musk was raised in South Africa.>
(Elon Musk was raised in South Africa.)
```

**Left padding augmented examples**

```
<space>Elon Musk was raised in South Africa.
<space><space>Elon Musk was raised in South Africa.
<tab>Elon Musk was raised in South Africa.
#<space>Elon Musk was raised in South Africa.
##<space>Elon Musk was raised in South Africa.
###<space>Elon Musk was raised in South Africa.
```

**Two augmented examples by our random space insertion described above**

---

### C.2  BIOGRAPHY DATA TEMPLATES

We include in the following example data templates for our synthesized biography dataset. For each biography profile, we include three rephrases of biography entries, 1 QA pair per attribute for instruction tuning, and another 5 QA pair per attribute for evaluation.

**Single document training data**

- Eden Benitez was born on January 18, 1959. He was from Santa Clarita. He graduated from University of Wisconsin, Madison. His major was Marketing. He worked for General Dynamics.

**Rephrase document training data**

- Eden Benitez was born on January 18, 1959. He was from Santa Clarita. He graduated from University of Wisconsin, Madison. His major was Marketing. He worked for General Dynamics.

- Eden Benitez completed his education at University of Wisconsin, Madison. His field of study was Marketing. He was employed at General Dynamics. His place of origin was Santa Clarita. He entered the world on January 18, 1959.

- Eden Benitez majored in Marketing. He developed his career at General Dynamics. His life began on January 18, 1959. He attended University of Wisconsin, Madison. He came from Santa Clarita.

**Instruction tuning QA pairs**

- When was Eden Benitez born? January 18, 1959
- Which university did Eden Benitez graduate from? University of Wisconsin, Madison
- Which company did Eden Benitez work for? General Dynamics
- Where was Eden Benitez from? Santa Clarita
- What was Eden Benitez's major?" Marketing

**Evaluation QA pairs**

- When did Eden Benitez come into this world? January 18, 1959
- What was Eden Benitez's birth date? January 18, 1959
- When was Eden Benitez brought into the world? January 18, 1959
- When did Eden Benitez first open his eyes? January 18, 1959
- What was the birth date of Eden Benitez? January 18, 1959
- Which university did Eden Benitez finish his education at? University of Wisconsin, Madison
- Which university did Eden Benitez complete his degree program at? University of Wisconsin, Madison
- Which university did Eden Benitez obtain his degree from? University of Wisconsin, Madison
- Which university did Eden Benitez receive education at? University of Wisconsin, Madison
- Which university did Eden Benitez earn his degree from? University of Wisconsin, Madison
- Which company did Eden Benitez have a job at? General Dynamics
- Which company did Eden Benitez find employment at? General Dynamics
- Which company did Eden Benitez work at? General Dynamics
- Which company did Eden Benitez have a professional role at? General Dynamics
- Which company did Eden Benitez hold a position at? General Dynamics
- Where was Eden Benitez's hometown? Santa Clarita
- Where did Eden Benitez originate from? Santa Clarita
- Where was Eden Benitez raised? Santa Clarita
- Where did Eden Benitez hail from? Santa Clarita
- Where was Eden Benitez a native of? Santa Clarita

- What major did Eden Benitez pursue a degree in? Marketing
- What major did Eden Benitez dedicate his studies to? Marketing
- What major did Eden Benitez work toward earning a degree in? Marketing
- What major did Eden Benitez study? Marketing
- What major was Eden Benitez majoring in? Marketing

