# OpenReview forum: "Effective LLM Knowledge Learning Requires Rethinking Generalization"
_ICLR.cc/2025/Conference — Submitted to ICLR 2025_

### Official Review · Reviewer_ywRf · 2024-11-03

**Soundness:** 3
**Presentation:** 2
**Contribution:** 2
**Rating:** 3
**Confidence:** 4

**Summary:**

This work suggests that knowledge acquisition for large language models (LLMs) can improve through strategies aimed at strengthening generalization in supervised tasks. The approach presented involves text paraphrasing in document format and the use of sharpness-aware minimization (SAM). Experimental results indicate that these methods support effective knowledge learning and can be adapted for instruction tuning. Additionally, training on paraphrased documents appears to facilitate knowledge extraction, an observation consistent with previous research.

**Strengths:**

- The study shows that knowledge learning performance can significantly improve through simple data augmentation, such as controlling spaces or adding special characters around sentences. This is an intriguing and noteworthy observation.
- The paper is easy to follow due to its clear writing.

**Weaknesses:**

- **Application of SAM**: Is there a particular reason SAM is expected to perform well in knowledge learning? From my reading, it seems this work merely applies SAM in a knowledge learning context without providing new insight into its specific relevance for knowledge learning.
- **Limitations of Observations**: As the authors mention, the effectiveness of rephrasing data has been previously reported (e.g., [1], [2]). Therefore, the observation in Section 3.3 is not novel, although it is valuable to validate their experimental setup.
- **Lack of Analysis**: There is insufficient analysis of the effectiveness of each data augmentation method. Additionally, comparing their approach to other paraphrasing methods, such as EDA or LLM-based paraphrasing, would clarify the unique advantages of their method.

References

[1] Allen-Zhu et al., Physics of language models: Part 3.1, knowledge storage and extraction

[2] Ovadia et al., Fine-tuning or retrieval? comparing knowledge injection in llms

**Questions:**

- In Lines 395–397 and 466–468, the authors suggest that performance improvements might not be due solely to extended training steps. To verify this, it would help to include a graph with training steps (not epochs) on the x-axis and accuracy on the y-axis, comparing performance with and without their method.
- Which data augmentation approach is most effective? Section 4.1 introduces various formatting variants, so an ablation study could clarify which variant contributes the most to performance.
- It would be beneficial to compare the proposed method with EDA or LLM-based paraphrasing. While the method presented here is impressively simple and effective, further validation is needed to establish it as the most effective approach. Although LLM-based paraphrasing may have reliability issues (as noted in Line 285), including an evaluation of it here would be informative.

---

> ### Author Response · Authors · 2024-11-20
>
> Thank you very much for taking the time to review our work and provide thoughtful feedback. Below, we have provided detailed responses to your questions. We hope these address your concerns and encourage you to consider increasing your score. Should you have any remaining questions or require further clarification, please do not hesitate to let us know, and we will do our best to clarify.
>
> **Q1: Limitations of Observations: As the authors mention, the effectiveness of rephrasing data has been previously reported (e.g., [1], [2]). Therefore, the observation in Section 3.3 is not novel, although it is valuable to validate their experimental setup.**
>
> We need to clarify that [1, 2] only make the empirical observation that training on paraphrased documents helps knowledge learning. They do not pose any understanding on why adding paraphrased documents is helpful.
>
> We propose a hypothesis that LLM knowledge learning is an implicit supervised learning problem. Assume the training document is ``<bos> Elon Musk was raised in South Africa`` and the corresponding question is ``<bos> Question: Where did Elon Musk grow up? Answer: South Africa``. In the training sentence, a **training sample** has the input ``<bos> Elon Musk was raised in`` and the label ``South``. In the corresponding question, a **testing sample** has the input ``<bos> Question: Where did Elon Musk grow up? Answer:`` and also has ``South`` as the label. If training on the **training sample** leads to the accuracy increase for the **testing sample**, it means that the **training sample** and the **testing sample** are from the same distribution and thus LLM knowledge learning is an implicit supervised learning problem. This might be not obvious from the human perspective, as the **training sample** and **testing sample** differ in structure: one is a declarative sentence, while the other is in the form of a question and answer. We conduct an experiment in Section 3.3 to verify our hypothesis. The results in Figure 1 in the main paper validate our hypothesis. Our hypothesis also explains the performance gain for adding paraphrased documents. Paraphrasing increases the number of in-distribution **training sample** and thus can lead to more accurate predictions on the **testing sample**. Results in Figure 1 in the main paper also validate this explanation. Therefore, our paper provides a systematic way of understanding LLM knowledge learning, while previous works [1,2] only have empirical observations but not explanations. Also, our experiment in Section 3.3 is conducted in the continued pre-training stage. This is different from [1, 2]'s empirical observation, which is made after the instruction tuning stage.
>
> Further, [1, 2] only studies the knowledge learning performance after the instruction tuning stage, while we analyze both the continued pre-training (Section 3.2, Section 3.3) and the instruction tuning stage (Section 4.3). The experiment in Table 2 of the main text also validates that our method is effective for both the continued pre-training and instruction tuning stages. For Qwen 2 1.5B, we can see that paraphrasing only achieves an exact match of 24.9 after continued pre-training. Our method achieves an exact match of 43.2 without paraphrased documents and an exact match of 74.9 with paraphrased documents, which means our method leads to effective knowledge learning and extraction even without the instruction tuning stage.  This result is entirely novel and different from naive paraphrasing used in prior studies [1, 2].
>
> We pose and verify that LLM knowledge learning is an implicit supervised learning problem. This provides a systematic way of understanding and improving LLM knowledge learning. Based on our understanding, we are able to propose our formatting-based augmentation and the application of SAM as the optimizer to improve LLM knowledge learning.
>
> According to your comment, we will add more explanation in the revised paper.
>
> [1] Physics of language models: Part 3.1, knowledge storage and extraction
>
> [2] Fine-Tuning or Retrieval? Comparing Knowledge Injection in LLMs

---

> ### Author Response · Authors · 2024-11-20
>
> **Q2: Application of SAM: Is there a particular reason SAM is expected to perform well in knowledge learning? From my reading, it seems this work merely applies SAM in a knowledge learning context without providing new insight into its specific relevance for knowledge learning.**
>
> In Section 3 of the main text and Q1 above, we demonstrate that LLM knowledge learning is implicitly a supervised learning problem. As SAM is designed to improve the generalization performance for supervised learning problems, we apply it to improve the generalization ability for knowledge learning. Our verification that LLM knowledge learning is implicitly a supervised learning problem is crucial for the application of SAM since it is unknown whether SAM is helpful for unsupervised learning on text corpus. Also, SAM's effectiveness highly relies on adequate in-distribution training samples, which are provided by our formatting-based data augmentation. Therefore, the combination of our formatting-based data augmentation and SAM leads to more considerable enhancement for LLM knowledge learning.
>
> From Table 4 in the main text, we can see that integrating data augmentation or SAM alone can already make a substantial improvement over naive continued pre-training. Integrating the combination of them leads to more significant performance gain.
>
>
> **Q3: Comparison with other paraphrasing methods, such as EDA or LLM-based paraphrasing.**
>
> We apologize for the lack of clarity in describing our experiments. Our paper has experiments comparing with paraphrasing. In the main text, we use **rephrase** to refer to training with paraphrased documents. Specifically, we paraphrase the biography templates using LLM and fill in the same biography profile attributes following [1]. We will improve the description in the revised paper more clearly.
>
> Quantitative results in Table 2 of the main text show that using our method (**single w/ ours**) significantly
> outperforms training on paraphrased samples (**rephrase**) for continued pre-training and leads to on-par or better performance for instruction tuning. For Qwen 2 1.5B, (**single w/ ours**) has 18\% and 5\% improvement in the exact match for continued pre-training and instruction tuning over (**rephrase**).
>
> Also, we want to clarify that our method is orthogonal to paraphrasing using LLM. From Table 2, we can see that applying our method on paraphrased samples (**rephrase w/ ours**) produces substantial enhancement over naive training on paraphrased samples (**rephrase**). For Qwen 2 1.5B, (**rephrase w/ ours**) has 50\% and 20\% improvement in the exact match for continued pre-training and instruction tuning over (**rephrase**). This further indicates the versatility of our method.
>
> In the table below, we also include the result of using EDA for training Qwen 2
> 1.5B on the biography dataset during the continued pre-training stage. It can be seen that EDA is harmful for knowledge learning. The exact match decreases from 7.1 to 0 when applying EDA. This is because EDA uses random word insertion, random word deletion, and random word swap, which are highly likely to alter the knowledge in documents. This further demonstrates our formatting-based augmentation's advantage that it reliably increases the number of in-distribution samples without changing the knowledge in documents.
>
> | Data Augmentation | EM | Recall | F1 |
> |----------|----------|----------|----------|
> | w/o data augmentation    | 7.1   | 16.1   | 12.4   |
> | w/ EDA    | 0.0   | 9.0   | 3.8   |
> | w/ our data augmentation    | 24.3   | 38.6   | 33.1   |
> | w/ paraphrasing    | 24.9   |  45.4   |  35.1   |
> | w/ paraphrasing + our data augmentation  | 54.9   |  78.8   | 65.3   |
>
> According to your suggestion, we will add the comparison with EDA in the revised paper.
>
> [1] Physics of language models: Part 3.1, knowledge storage and extraction
>
> [2] Fine-Tuning or Retrieval? Comparing Knowledge Injection in LLMs

---

> ### Author Response · Authors · 2024-11-20
>
> **Q4: In Lines 395–397 and 466–468, the authors suggest that performance improvements might not be due solely to extended training steps. To verify this, it would help to include a graph with training steps (not epochs) on the x-axis and accuracy on the y-axis, comparing performance with and without their method.**
>
> We need to clarify that all methods are trained for the same number of iterations in our experiments to ensure a fair
> comparison. In Lines 395–397 of the main paper, we state that all methods, regardless of data augmentation or
> the addition of paraphrased texts, are training for the same number of iterations as training without rephrased samples. The epoch in the x-axis of Figure 1 in the main text actually means the epoch count for training without rephrased samples. We will change the x-axis of Figure 1 to iteration to make it more clear.
>
> Thus, no methods are gaining an advantage by training using more iterations or having more passes of data.
>
> Additionally, we follow your suggestions to compare the average first knowledge token accuracy conditioned on context questions for models continually pre-trained with and without our method in the plot added in Section A.3. The experiment setting is the same as in Section 3.3 of the main text. We can see that our method leads to significant improvement over naive continued pre-training with and without rephrased samples.
>
>
> **Q5: Which data augmentation approach is most effective? Section 4.1 introduces various formatting variants, so an ablation study could clarify which variant contributes the most to performance.**
>
> Thank you for your suggestions. In the table below, We include the ablation study on our proposed three formatting-based augmentations for Qwen 2 1.5B on the biography dataset during the continued pre-training stage. The results are summarized in the table below. It can be seen that all three types of formatting-based augmentation lead to significant improvement. Among them, Wrapping and Random space insertion are more effective than Left padding. The combination of all three types of augmentation leads to more considerable enhancement.
> | Data Augmentation | EM | Recall | F1 |
> |----------|----------|----------|----------|
> | w/o data augmentation    | 7.1   | 16.1   | 12.4   |
> | w/ Wrapping    | 21.0   | 34.0   | 28.3   |
> | w/ Left padding    | 10.7   | 24.1   | 16.9   |
> | w/ Random space insertion    | 15.2   | 37.7   | 24.7   |
> | w/ all three data augmentation    | 24.3   | 38.6   | 33.1   |
>
> Thanks to your suggestion, we will add the ablation study for our formatting-based data augmentation to the revised paper.

---

> ### Author Response · Authors · 2024-11-24
>
> Dear Reviewer ywRf,
>
> Thank you for your detailed review and insightful feedback. We have carefully addressed your concerns in our rebuttal and would greatly value your engagement during the rebuttal period to assess whether our responses adequately address your comments. Your expertise is highly appreciated, and we are more than happy to provide any additional clarification or further details if needed.
>
> Thank you once again for your time and thoughtful review.
>
> Best regards

---

> > ### Comment · Reviewer_ywRf · 2024-11-25
> >
> > Thank you for your detailed response and the efforts invested during the rebuttal phase.
> > I appreciate the additional experiments provided in response to Q3, Q4, and Appendix Figure 2, which address some of my concerns about the experimental setup. Therefore, I raise my score for soundness from 2 to 3.
> > However, I still have concerns regarding the contribution of this work for several reasons.
> >
> > First, framing the knowledge learning problem as an implicit supervised learning task appears to oversimplify the complexities involved. For example, consider the sentence “Elon Musk was raised in South Africa.”
> > Training a language model on this sentence using next-token prediction loss might be interpreted as implicitly maximizing the likelihood of predicting “South” given the context “Elon Musk was raised in.” However, in real-world scenarios, the pre-training corpus might only contain an alternative phrasing, such as “South Africa is where Elon Musk was raised.” In such cases, the model cannot be regarded as implicit supervised training for the specific question “Where did Elon Musk grow up?”
> > The inherent variability in phrasing within pre-training corpora challenges the notion that such training fully embodies supervised learning. To address this issue, more structured rephrasing or sentence reordering—beyond simple wrapping, left padding, or random space insertion—would be necessary, as supported by prior work [1,2].
> >
> > Second, the data augmentation method through formatting seems to represent merely a reformulation or paraphrasing of the input text. While the experimental results suggest that formatting changes improve the generalization of knowledge learning, the underlying mechanism remains unclear. For instance, rephrasing the sentence “Elon Musk was raised in South Africa” as “The place where Elon Musk grew up is South Africa” might enhance the model's ability to answer the question “Where did Elon Musk grow up?” However, it is not evident why such simple formatting adjustments would lead to better generalization in knowledge learning. The paper lacks compelling explanations or evidence to justify this phenomenon.
> >
> > Finally, the contribution of applying Sharpness-Aware Minimization (SAM) to the problem of knowledge learning remains unconvincing. Based on the response to Q2, the application of SAM appears to be a straightforward extension rather than a significant contribution. Demonstrating the benefits of SAM in the context of unsupervised learning on text corpora does not, in my view, constitute a substantial contribution.
> >
> > Providing additional experimental evidence and more robust explanations for these three issues would significantly strengthen this work in future revisions.
> >
> > References
> >
> > [1] *The Reversal Curse: LLMs trained on "A is B" fail to learn "B is A”*
> >
> > [2] *Reverse Training to Nurse the Reversal Curse*

---

> > > ### Author Response · Authors · 2024-11-26
> > >
> > > We would like to thank the reviewer ywRf for their detailed feedback. We appreciate the recognition of the additional experiments provided in response to Q3, Q4, and Appendix Figure 2, and we are pleased that these address several of your concerns regarding the experimental setup. As noted, we understand that the soundness of the work has been rated more favorably, and we hope the following clarifications will further address the remaining points of concern.
> > >
> > > - **Addressing concerns in paragraph 2:**
> > >
> > > We respectfully disagree with the reviewer’s interpretation of our proposed implicit supervised learning perspective in the context of knowledge learning.
> > >
> > > **<1>** The issue described by the reviewer is indeed the reversal curse problem [1-5]. Contrary to the reviewer's suggestion, when the training corpus only has "South Africa is where Elon Musk was raised.", the autoregressively trained model can **never** answer questions like "Where did Elon Musk grow up?". Even when training with paraphrased documents, the accuracy for reversal QA **always remains 0** [1, 2]. Our response to the Q1 of Reviewer 1bPQ also explains this from our proposed perspective. Therefore, the reversal curse is considered as a distinct research problem [1-5] by the community. The reversal curse problem falls outside the scope of this paper. Our goal is not to address the reversal curse but to advance the understanding of knowledge learning and improve it.
> > >
> > > **<2>** Our proposed perspective to view knowledge learning as an implicitly supervised problem is thoroughly validated in Section 3.3. The methods we developed based on our perspective all lead to substantial improvement in knowledge learning, as shown in our experiments. These results further validate the correctness of our perspective that knowledge learning is implicitly supervised.
> > >
> > > **<3>** The approaches aimed at mitigating the reversal curse [2, 4] all involve reversing parts or entire training sentences. When such reversed sentences are incorporated into the training process, our proposed perspective can effectively accommodate this scenario, and our methods would still be effective.
> > >
> > > [1] The Reversal Curse: LLMs trained on "A is B" fail to learn "B is A"
> > >
> > > [2] Reverse Training to Nurse the Reversal Curse
> > >
> > > [3] Physics of Language Models: Part 3.2, Knowledge Manipulation
> > >
> > > [4] Rethinking the Reversal Curse of LLMs: a Prescription from Human Knowledge Reversal
> > >
> > > [5] Reverse Modeling in Large Language Models

---

> > > ### Author Response · Authors · 2024-11-26
> > >
> > > - **Addressing concerns in paragraph 3:**
> > >
> > > Again, we respectfully disagree with the reviewer’s assessment of the data augmentation method used in our work.
> > >
> > > **<1>** **Before our proposed perspective to view knowledge learning as an implicitly supervised problem, there is no systematic understanding of the underlying mechanism of paraphrasing either.** Previous works [1, 2] only have empirical observations that paraphrasing can help. Our work fills this gap by offering a framework that systematically explains why presenting documents in diverse formats and wording enhances the generalization of knowledge learning.
> > >
> > > **<2>** Intuitively, our formatting-based data augmentation presents the training document in diverse formats. It can increase the data diversity just like paraphrasing, which presents the training document in different wording. Therefore, our formatting-based data augmentation should be helpful from the intuitive perspective.
> > >
> > > **<3>** Formally, we have demonstrated that paraphrased documents and the original document are from the same distribution, and training with paraphrased documents can improve QA accuracy. Presenting the original document in diverse formats, like wrapping the original document in quotation marks, creates augmented documents that are more similar and closer to the original document, than paraphrased ones. Thus, our formatting-based data augmentation is guaranteed to produce in-distribution samples, which help the generalization of knowledge learning based on our proposed perspective to view knowledge learning as an implicitly supervised problem. Also, our formatting-based data augmentation can easily and automatically create a lot more in-distribution samples without altering the facts in original documents, which is difficult to achieve using paraphrasing. We note again that without paraphrased samples, LLM knowledge learning is a **1-shot** supervised learning problem. With paraphrased samples, LLM knowledge learning is a **few-shot** supervised learning problem. Our formatting-based data augmentation further alleviates the data scarcity in knowledge learning and improves performance. Extensive experiments in the paper and our previous response validate the effectiveness of our formatting-based data augmentation.
> > >
> > > **<4>** The rigorous understanding of the underlying mechanism of data augmentation is beyond the scope of this paper, which is usually discussed in theory papers [3, 4].
> > > - **Addressing concerns in paragraph 4:**
> > >
> > > We respectfully disagree with the reviewer’s critique of our application of Sharpness-Aware Minimization (SAM). The integration of SAM as the optimizer is based on our novel perspective that knowledge learning is implicitly a supervised problem. The contribution is that we pose and verify our perspective. The integration of SAM would be **baseless** without our perspective. Another contribution of integrating SAM is it significantly boosts the performance, as demonstrated in our experiments. As we have clearly demonstrated knowledge learning is implicitly a supervised problem, demonstrating SAM's benefit for unsupervised problems is unrelated and beyond the focus of this paper.
> > >
> > > [1] Physics of language models: Part 3.1, knowledge storage and extraction
> > >
> > > [2] Fine-Tuning or Retrieval? Comparing Knowledge Injection in LLMs
> > >
> > > [3] A Kernel Theory of Modern Data Augmentation
> > >
> > > [4] A Group-Theoretic Framework for Data Augmentation

---

> > > ### Author Response · Authors · 2024-11-29
> > >
> > > Dear Reviewer ywRf
> > >
> > > Thank you again for your thoughtful feedback and for recognizing the additional experiments and explanations we provided. We appreciate your effort in engaging with our work and understand the concerns you raised.
> > >
> > > We would like to further clarify your assumption that training on a sentence like “South Africa is where Elon Musk was raised” would enable the model to predict “South Africa” for the question “Where did Elon Musk grow up?.” This assumption overlooks the nature of the reversal curse. We illustrated in previous responses that there have been works empirically demonstrated that training on forwardly phrased knowledge (e.g., "South Africa is where Elon Musk was raised") results in models **having 0 accuracy** for reverse questions (e.g., “Where did Elon Musk grow up?”) [1, 2].
> > >
> > > We want to add that recent theoretical work [3] has formalized this phenomenon, proving that training a model to predict the next token in forward directions makes it almost impossible to successfully predict the reverse direction： "Theorem 3 shows that although the direction presented in the training set can be learned nearly perfectly, the model’s next token prediction of the reverse direction is almost a random guess." Therefore, researchers usually consider the reversal curse a standalone problem rather than standard knowledge learning. Thus, our framing of the knowledge learning problem as an implicit supervised learning task does not oversimplify the complexities.
> > >
> > > We hope this explanation helps clarify the issue and addresses your concerns.
> > >
> > > Authors
> > >
> > > [1] The Reversal Curse: LLMs trained on "A is B" fail to learn "B is A"
> > >
> > > [2] Reverse Training to Nurse the Reversal Curse
> > >
> > > [3] Towards a Theoretical Understanding of the ‘Reversal Curse’ via Training Dynamics

---

### Official Review · Reviewer_kH5h · 2024-11-04

**Soundness:** 2
**Presentation:** 3
**Contribution:** 3
**Rating:** 6
**Confidence:** 2

**Summary:**

This paper explores how LLMs acquire and retrieve knowledge through autoregressive pre-training. The authors found that knowledge learning for LLMs is an implicitly supervised problem. This observation is valuable. They propose a data augmentation method to increase in-distribution samples and introduce sharpness-aware minimization as an optimizer. Experiments validate the supervised nature of LLM knowledge learning and demonstrate the effectiveness of the proposed methods.

**Strengths:**

1.I think this is an worth-investigating topic. It is still not understood how knowledge is acquired via autoregressive pre-training. This lack of understanding greatly hinders effective LLM knowledge learning. The paper provides insights into how LLMs acquire knowledge through auto-regressive pre-training. The authors verify that knowledge learning for LLMs is an implicitly supervised problem, which is a novel finding.

2.They propose a data augmentation method and introduce sharpness-aware minimization as an optimizer to improve knowledge acquisition.

3.The analysis is extended to the instruction tuning phase, highlighting the importance of generalization on different questions with the same answer.

4. Extensive experiments and ablation studies validate the findings and demonstrate the effectiveness of the proposed methods.

**Weaknesses:**

1.In Figure 1, it appears that the text within the figure is excessively large in relation to the size of its corresponding captions.

2.What would happen if the number of tokens remained constant in the data augmentation experiment? If it performs worse than before, does it mean that a decrease in the overall knowledge contained will lead to poor knowledge acquisition?

**Questions:**

See Weaknesses.

---

> ### Author Response · Authors · 2024-11-20
>
> Thank you very much for taking the time to evaluate our paper and providing your positive comments. We are delighted that you recognize that our work contributes to a worth-investigating topic in the research field. We appreciate the opportunity to address the points raised in your review and provide further clarification.
>
>
>
> **Q1: Size of figure 1.**
>
> Thank you for this suggestion! We will make Figure 1 smaller so that the text within the figure is relatively the same size as the captions.
>
> **Q2: What would happen if the number of tokens remained constant in the data augmentation experiment? If it performs worse than before, does it mean that a decrease in the overall knowledge contained will lead to poor knowledge acquisition?**
>
> In Lines 395–397 of the main text, we state that in our experiment all methods are training for the same number of iterations, regardless of data augmentation or the addition of paraphrased texts. In this way, the same number of training documents are passed in training for all methods. For the number of tokens in a single training document, it can be seen in Section 4.1 that our formatting-based data augmentation only adds a marginal number of tokens to training documents. Therefore, the number of tokens trained with and without our data augmentation is almost the same. Thus, the knowledge learning performance with the number of tokens remaining constant would be similar to our reported ones.

---

> > ### Comment · Reviewer_kH5h · 2024-11-22
> > **Official Comment**
> >
> > Thank you for the clarifications. I will keep my positive score.

---

### Official Review · Reviewer_1bPQ · 2024-11-05

**Soundness:** 3
**Presentation:** 3
**Contribution:** 3
**Rating:** 5
**Confidence:** 4

**Summary:**

This paper explores how LLMs acquire and generalize knowledge through training, addressing the previously unclear process of knowledge acquisition in autoregressive pre-training and its extraction in question-answering. The authors reveal that knowledge learning in LLMs functions as an implicit supervised task embedded within the autoregressive pre-training objective. Their key findings are:
1. They propose a way to generate in-distribution training samples by diverse document formatting. This automatic augmentation method mitigates the risk of altering facts in documents.
2.  They verify the hypothesis that that training documents and knowledge-based questions align in distribution, making knowledge learning a supervised problem.

**Strengths:**

1. The authors proposed to apply Sharpness-Aware Minimization and document formatting-based data augmentation, the authors provide practical methods to improve LLM generalization on knowledge learning.

2. The paper introduces an new perspective by framing knowledge learning in LLMs as an implicit supervised task.

3. The paper is well-structured and clearly written,

**Weaknesses:**

1. Comparison with Paraphrasing: While the authors propose formatting-based data augmentation to prevent factual alterations, it would be useful to include a direct comparison with paraphrasing, as it remains a widely used method. A quantitative analysis would help clarify if the formatting approach is comparably effective or if paraphrasing has advantages under certain conditions. Even if paraphrasing risks altering facts, seeing how the two methods differ in terms of model performance could demonstrate the benefits and limitations of each. Could an ablation study be conducted to compare their formatting-based augmentation against paraphrasing on a subset of data where paraphrasing is safe? This would provide quantitative evidence of the relative effectiveness of both approaches.

2. The significant of the finding is not clear, especially relative to established concepts like the “reversal curse” and related works is well taken. Both [1] and [2] delve into the issue of the knowledge learning problem and found that training on one single knowledge statement is not enough for the model to capture the knowldege. Differentiating their contribution more clearly would be helpful. Could you explain how the framing of knowledge learning as a supervised task either differs from or builds upon the insights in these prior works? It would be useful to have a dedicated paragraphs or evaluation to directly compare your findings with [1] and [2]

3.  The adaptation to instruction tuning may not provide a strong motivation for using the proposed method, as instruction tuning primarily focuses on learning the format rather than acquiring knowledge. Could you provide empirical evidence showing how your method impacts knowledge acquisition during instruction tuning, beyond just format learning?

[1] The Reversal Curse: LLMs trained on "A is B" fail to learn "B is A"
[2] Physics of Language Models: Part 3.1, Knowledge Storage and Extraction

**Questions:**

Why does the combination of SAM and data augmentation provide such a significant improvement, whereas using only one of these methods does not?

---

> ### Author Response · Authors · 2024-11-20
>
> Thank you for taking the time to review our work and provide thoughtful feedback. Below, we have provided detailed responses to your questions. We hope these address your concerns and encourage you to consider increasing your score.
>
> **Q1: The significance of our finding and comparison to existing works.**
>
> Thank you for your suggestions. We will add more discussion comparing our findings with [1, 2] in our revision.
>
> Below are our responses to Q1:
>
> - **Comparison with Physics of Language Models: Part 3.1**
>
> [1] only makes the empirical observation that training on paraphrased documents helps knowledge learning. They do not pose any understanding on why adding paraphrased documents is helpful.
>
> We propose a hypothesis that LLM knowledge learning is an implicit supervised learning problem. Assume the training document is ``<bos> Elon Musk was raised in South Africa`` and the corresponding question is ``<bos> Question: Where did Elon Musk grow up? Answer: South Africa``. In the training sentence, a **training sample** has the input ``<bos> Elon Musk was raised in`` and the label ``South``. In the corresponding question, a **testing sample** has the input ``<bos> Question: Where did Elon Musk grow up? Answer:`` and also has ``South`` as the label. If training on the **training sample** leads to the accuracy increase for the **testing sample**, it means that the **training sample** and the **testing sample** are from the same distribution and thus LLM knowledge learning is an implicit supervised learning problem. This might be not obvious from the human perspective, as the **training sample** and **testing sample** differ in structure: one is a declarative sentence, while the other is in the form of QA. We conduct an experiment in Section 3.3 to verify our hypothesis. Results in Figure 1 in the main paper validate our hypothesis. Our hypothesis also explains the performance gain for adding paraphrased documents. Paraphrasing increases the number of in-distribution **training sample** and thus leads to more accurate predictions on the **testing sample**. Results in Figure 1 also validate this explanation. Therefore, our paper provides a systematic way of understanding LLM knowledge learning, while previous works [1] only have empirical observations but not explanations. Also, our experiment in Section 3.3 is conducted in the continued pre-training stage. This is different from [1]'s empirical observation, which is made after the instruction tuning stage.
>
> Further, [1] only studies the knowledge learning performance after the instruction tuning stage, while we analyze both the continued pre-training (Section 3.2, Section 3.3) and the instruction tuning stage (Section 4.3). The experiment in Table 2 of the main text also validates that our method is effective for both the continued pre-training and instruction tuning stages. For Qwen 2 1.5B, we can see that paraphrasing only achieves an exact match of 24.9 after continued pre-training. Our method achieves an exact match of 43.2 without paraphrased documents and an exact match of 74.9 with paraphrased documents, which means our method leads to effective knowledge learning and extraction even without the instruction tuning stage. This result is entirely novel and different from naive paraphrasing used in prior studies [1].
>
> We pose and verify that LLM knowledge learning is an implicit supervised learning problem. This provides a systematic way of understanding and improving LLM knowledge learning. Based on our understanding, we are able to propose our formatting-based augmentation and the application of SAM as the optimizer.
>
> - **Comparison with The Reversal Curse**
>
> The reversal curse is an empirical observation that LLMs trained on "A is B" fail to learn "B is A" [2]. We do not aim to alleviate the reversal curse problem in this paper, however, our proposed perspective to view LLM knowledge learning as a supervised learning problem can explain the existence of the reversal curse.
>
> Assume the training sentence is ``<bos> A is B`` and its reverse sentence is ``<bos> B is A``, where \<bos\> is the special begin-of-sentence token indicating the beginning of a sentence. During training, the input ``<bos>`` has ``A`` as the label. In the reverse sentence, the input ``<bos> B is`` has ``A`` as the label. ``<bos>`` and ``<bos> B is`` are too different and are impossible to be from the same distribution. Although sharing ``A`` as the label, training on ``<bos>`` cannot generalize to ``<bos> B is``. This explains the empirical observation of the reversal curse.
>
> This also corresponds to the empirical observation that paraphrasing the training sentence ``<bos> A is B`` cannot alleviate the reversal curse [2], and only reverse parts or the whole training sentence can help [3].
>
> [1] Physics of Language Models: Part 3.1, Knowledge Storage and Extraction
>
> [2] The Reversal Curse: LLMs trained on "A is B" fail to learn "B is A"
>
> [3] Reverse Training to Nurse the Reversal Curse

---

> ### Author Response · Authors · 2024-11-20
>
> **Q2: Comparison with paraphrasing.**
>
> We apologize for the lack of clarity in describing our experiments. Our paper has experiments comparing with paraphrasing. In the main text, we use **rephrase** to refer to training with paraphrased documents. Specifically, we paraphrase the biography templates using LLM and fill in the same biography profile attributes following [1]. We will improve the description in the revised paper.
>
> Quantitative results in Table 2 of the main text show that using our method (**single w/ ours**) significantly
> outperforms training on paraphrased samples (**rephrase**) for continued pre-training and leads to on-par or better performance for instruction tuning. For Qwen 2 1.5B, (**single w/ ours**) has 18\% and 5\% improvement in the exact match for continued pre-training and instruction tuning over (**rephrase**).
>
> Also, we want to clarify that our method is orthogonal to paraphrasing using LLM. From Table 2, we can see that applying our method on paraphrased samples (**rephrase w/ ours**) produces substantial enhancement over naive training on paraphrased samples (**rephrase**). For Qwen 2 1.5B, (**rephrase w/ ours**) has 50\% and 20\% improvement in the exact match for continued pre-training and instruction tuning over (**rephrase**). This further indicates the versatility of our method.
>
> **Q3: Motivation for adaptation to instruction tuning.**
>
> We appreciate for inquiring about the motivation. It is true that knowledge is learned during the continued pre-training stage. Instruction tuning makes the knowledge extractable in the QA format. However, as we analyzed in Section 4.3, different users might ask for the same knowledge using different question prompts. It is important for LLMs to stably output the learned knowledge given varied question prompts that might not be seen during the instruction tuning stage. Therefore, we should also consider generalization to different question prompts asking for the same knowledge in the instruction tuning stage. Thus, we propose to apply our formatting-based data augmentation to question prompts of the instruction tuning training samples and apply SAM as the optimizer during the instruction tuning stage. Quantitative results in Table 5 of the main paper also validate our motivation. Applying formatting-based augmentation or SAM alone can improve the exact match by around 3\% and applying both can improve the exact match by 5\%, which is a significant improvement.
>
> According to your comment, we will add more explanation in the revised paper.
>
> **Q4: Why does the combination of SAM and data augmentation provide such a significant improvement, whereas using only one of these methods does not?**
>
> Your questions indeed touch on the contributions of our work. Thank you so much!
>
> In Section 3 of the main text and Q2 above, we demonstrate that LLM knowledge learning is implicitly a supervised learning problem. Without paraphrased samples, LLM knowledge learning is a **1-shot** supervised learning problem. With paraphrased samples, LLM knowledge learning is a **few-shot** supervised learning problem. The absence of adequate in-distribution training samples makes LLM knowledge learning extremely difficult.
>
> Our formatting-based data augmentation effectively increases the number of in-distribution samples. SAM, on the other hand, can improve the generalization performance for supervised learning problems. SAM's effectiveness highly relies on adequate in-distribution training samples, which are provided by our formatting-based data augmentation. Therefore, the combination of our formatting-based data augmentation and SAM leads to more considerable enhancement for LLM knowledge learning.
>
> From Table 4 in the main text, we can see that integrating data augmentation or SAM alone can already make a substantial improvement over naive continued pre-training. Integrating the combination of them leads to more significant performance gain.
>
>
> According to your questions, we will add more explanation in the revised paper. Thank you so much again!
>
> [1] Physics of Language Models: Part 3.1, Knowledge Storage and Extraction
>
> [2] The Reversal Curse: LLMs trained on "A is B" fail to learn "B is A"
>
> [3] Reverse Training to Nurse the Reversal Curse

---

> ### Author Response · Authors · 2024-11-24
>
> Dear Reviewer 1bPQ,
>
> Thank you for your detailed review and insightful feedback. We have carefully addressed your concerns in our rebuttal and would greatly value your engagement during the rebuttal period to assess whether our responses adequately address your comments. Your expertise is highly appreciated, and we are more than happy to provide any additional clarification or further details if needed.
>
> Thank you once again for your time and thoughtful review.
>
> Best regards

---

> > ### Comment · Reviewer_1bPQ · 2024-11-26
> >
> > Thanks for your detailed response. I increased my scores.

---

> > > ### Author Response · Authors · 2024-11-26
> > >
> > > Dear Reviewer 1bPQ,
> > >
> > > Thank you for your thoughtful feedback and for acknowledging the detailed responses in our rebuttal. We sincerely appreciate the increased score, as it reflects that we have addressed some of your initial concerns.
> > >
> > > We understand that your current assessment suggests there may still be concerns or areas of improvement. If there are specific remaining issues or aspects of the paper that you feel could be improved or clarified further, we would be more than happy to address them during this discussion phase.
> > >
> > > We are confident that our paper offers a substantial contribution by systematically bridging the understanding gap in LLM knowledge learning and proposing two highly effective methods grounded in our novel insights. We would greatly appreciate any further guidance or suggestions that could help strengthen your confidence in our contributions.
> > >
> > > Thank you again for your time and constructive feedback. We deeply value the opportunity to refine our work through this discussion and welcome any additional insights or suggestions you might have.
> > >
> > > Best regards,
> > >
> > > Authors

---

### Author Response · Authors · 2024-11-24
**Summary of Revisions**

The following updates have been incorporated into the draft based on the insightful feedback from the reviewers:

- **Addressing concerns of Reviewer 1bPQ and Reviewer ywRf:**

We expanded discussions on prior works in the main paper and Section B of the appendix. Additionally, the overall narrative has been refined for improved clarity and coherence.

- **Suggestions from Reviewer ywRf:**
    - An ablation study of our formatting-based augmentation has been included in Section A.1. Results indicate that the combination of the three proposed augmentations generates more diverse in-distribution samples and significantly enhances performance.
    - we compare our formatting-based augmentation with EDA in Section A.2. EDA would modify facts in documents, while our method avoids this issue. Our method significantly outperforms EDA.
    - A curve depicting the average first knowledge token accuracy, conditioned on context questions with and without our method, is now included in Section A.3. This demonstrates the substantial improvements in knowledge acquisition achieved with our approach, both with and without paraphrased documents.
- **Suggestions from Reviewer kH5h:**

We adjusted the size of Figure 1 to ensure the text within the figure is consistent in size with the captions.

---

### Meta-Review · Area_Chair_tnos · 2024-12-21

**Metareview:**

This paper explores how LLMs acquire and retrieve knowledge during autoregressive pre-training. The authors propose a method for generating in-distribution training samples using diverse document formatting to mitigate the risk of factual distortion. They also investigate the hypothesis that training documents and knowledge-based questions are aligned in distribution, framing knowledge learning as an implicitly supervised problem.

This work addresses the important open question of how LLMs acquire knowledge during autoregressive pre-training, offering valuable insights into this complex process. A key contribution is the observation that knowledge learning in LLMs can be viewed as an implicitly supervised task. Building on this insight, the authors propose a data augmentation method combined with sharpness-aware minimization to improve knowledge acquisition during pre-training. Their analysis extends to the instruction tuning phase, highlighting the importance of generalization across diverse question phrasings for the same underlying answer.

**Additional Comments On Reviewer Discussion:**

Reviewers raised concerns regarding comparisons with paraphrasing, the novelty of the approach, and the motivation for certain design choices. The authors have addressed most of these concerns. However, there appears to be a lack of strong consensus among the reviewers for acceptance, potentially due to perceived limitations in the significance of the findings.

Thanks for flagging ywRf. This decision has taken into account this.

---

### Decision · Program_Chairs · 2025-01-22

Reject